# In Silico Assessment of Potential Geroprotectors: From Separate Endpoints to Complex Pharmacotherapeutic Effects

**DOI:** 10.3390/ijms26188858

**Published:** 2025-09-11

**Authors:** Leonid Stolbov, Anastasia Rudik, Alexey Lagunin, Dmitry Druzhilovskiy, Dmitry Filimonov, Vladimir Poroikov

**Affiliations:** 1Institute of Biomedical Chemistry, 10 Bldg. 8, Pogodinskaya Str., 119121 Moscow, Russiavladimir.poroikov@ibmc.msk.ru (V.P.); 2Bioinformatics Department, Pirogov Russian National Research Medical University, 1 Bldg. 6 Ostrovityanova Str., 117513 Moscow, Russia

**Keywords:** aging, geroprotectors, in silico assessment, structure-activity relationships, PASS GERO

## Abstract

This study presents an approach for the in silico assessment of potential geroprotectors that target the multifaceted mechanisms of aging, implemented in the PASS GERO web application. This work is timely given the societal impact of aging—the primary risk factor for major chronic diseases. The urgent need to extend healthspan—the period of life spent in good health—motivates the search for compounds that modulate fundamental aging mechanisms. The model estimates the probabilities of 117 aging-related biological activities with high predictive accuracy, achieving an average Invariant Accuracy of Prediction (IAP) of 0.967 under cross-validation. Validation using known geroprotectors (rapamycin, metformin, and resveratrol) demonstrated strong concordance between predicted activities and documented molecular mechanisms of action. For instance, the model correctly predicted rapamycin’s inhibition of mTOR and metformin’s activation of AMPK. The PASS GERO web application provides a systematic strategy to prioritize novel compound candidates for experimental evaluation in anti-aging research. We discuss challenges including the chemical diversity of the training data, the need for validated biomarkers, and the limitations of translating computational predictions into clinical outcomes, positioning the tool as robust application for activity profiling in discovery workflows.

## 1. Introduction

Aging is a complex biological phenomenon characterized by the progressive decline of physiological resilience, leading to increased susceptibility to chronic diseases, functional impairment, and mortality [1,2]. This process involves interrelated molecular and cellular effects such as genomic instability [3], telomere shortening, epigenetic dysregulation, mitochondrial dysfunction, cellular senescence, and impaired nutrient-sensing pathways, being also influenced by lifestyle, genetics, and environmental factors. The above-mentioned mechanisms collectively lead to systemic dysfunction across tissue and organ systems [4,5].

With advancing age, the probability of chronic conditions including cardiovascular diseases, type 2 diabetes, cancer, and neurodegenerative disorders rises significantly [6]. While conventional medical approaches focus on treating these pathologies individually, they often address symptoms rather than targeting the underlying complex biological drivers of aging itself [7]. This reactive strategy fails to address the issues that lead to many age-related diseases simultaneously [8,9].

Geroprotectors—compounds designed to modulate the known aging pathways—are increasingly discussed in scientific literature as a transformative therapeutic strategy. Unlike single-disease therapies, geroprotectors aim to extend healthspan (the chronic disease-free period of life) and compress morbidity by intervening in fundamental aging mechanisms and thus increasing lifespan [10]. The conceptual shift towards geroprotection is based on the recognition of aging as a malleable biological process. Several studies in model organisms, from Caenorhabditis elegans to mice, have demonstrated that such interventions as caloric restriction, genetic modifications, and pharmacological agents (e.g., rapamycin) can extend lifespan. Translating these results to humans, however, remains challenging [11,12]. The described factors of aging complicate the identification of robust biomarkers and therapeutic targets. Moreover, the high costs, ethical constraints, and timelines of clinical trials for aging slow down such investigations. Despite these hurdles, geroprotective candidates like senolytics, mTOR inhibitors, NAD+ boosters, and metformin have emerged as front-runners, showing promise in preclinical studies and early-stage human trials [13,14]. Preclinical and early clinical studies suggest these agents may delay or prevent multiple age-related conditions, offering a systemic and prophylactic approach [10,15].

The statistics of publications dedicated to geroprotectors and presented in PubMed [16] demonstrate the exponential increase in the recent years (Figure 1).

The heterogeneity of aging processes demands interventions embracing multiple pathways at a time. Effective geroprotection requires polypharmacological actions, as aging itself arises from the interplay of diverse biological processes. Both the need to consider the complexity and heterogeneity of aging and the aim to intervene in several biological processes simultaneously make the discovery of potential geroprotectors particularly challenging. So, multiparametric estimations are required to capture the diverse aspects of aging [17,18].

Taking the complexity of aging process into account, the identification and evaluation of geroprotectors strictly rely on computational methods, particularly for virtual screening and multi-target activity estimation [19,20,21,22]. In silico approaches facilitate the prioritization of candidates by analyzing chemical structures, predicting biological activities, and modeling interactions with aging-related molecular targets. These tools address challenges posed by the chemical diversity of geroprotectors and the need for multi-objective optimization [23].

“One of the earliest and most widely used examples of data-mining target elucidation is the continuously curated and expanded Prediction of Activity Spectra for Substances (PASS) software” [24]. The PASS software estimates the probable biological activity profile corresponding to the structural formula of a drug-like organic compound. Classification of a compound’s belonging to the “actives” or “inactives” is based on the analysis of structure-activity relationships for the training set of compounds with the known biological activities using a machine learning approach. Particular compound’s biological activity is defined qualitatively as “active” or “inactive”. Molecular structure is described as Multilevel Neighborhoods of Atoms (MNA) descriptors. The algorithm of biological activity spectrum prediction is based on the improved naïve Bayes classifier. The current version of PASS, PASS 2024, estimates 9047 types of biological activity with an average invariant accuracy of 0.934 based on the training set that includes more than 1,482,930 drugs, drug-candidates, pharmaceutical agents, and chemical probes, as well as the compounds with known specific toxicity profiles. Hundreds of biological activities predicted by PASS 2024 may be considered eligible for the evaluation of potential geroprotectors, while some anti-aging activities should still be added to the existing SAR knowledgebase. Nevertheless, some applications of PASS for the evaluation of potential geroprotectors were already performed earlier [25,26,27].

This study presents the development and validation of a novel computational approach for in silico assessment of potential geroprotectors, implemented in a specialized version of our software, PASS GERO. We expanded the system with a comprehensive set of aging-related biological activities, yielding highly predictive models for 117 anti-aging mechanisms, including senolysis, mitochondrial function, and epigenetic regulation. These models showed strong performance, with a mean Invariant Accuracy of Prediction (IAP) of 0.967 in leave-one-out cross-validation and 0.966 in 20-fold cross-validation.

Importantly, applying PASS GERO to well-established geroprotectors such as rapamycin, metformin, and resveratrol produced results that closely matched their known mechanisms of action. For example, the model correctly predicted that rapamycin inhibits mTOR and enhances autophagy, and that resveratrol activates sirtuins, underscoring its biological relevance and its ability to recapitulate the complex polypharmacology of aging interventions.

The principle behind PASS GERO is straightforward: a researcher inputs a compound’s chemical structure—by drawing it or importing a structure file—and the application generates a comprehensive report predicting its activity across all 117 modeled mechanisms of aging. This delivers a multi-target polypharmacological profile in a single step, enabling the prioritization of candidates for subsequent research, including virtual screening of large libraries, guiding the synthesis of novel analogs, and planning targeted experimental studies.

PASS GERO not only provides a robust framework for predicting anti-aging activities but also delivers experimentally verifiable insights, making it a practical tool for prioritizing candidate compounds for further studies.

## 2. Results

### 2.1. Activities

We prepared the list of anti-aging biological activities based on their relevance to impact the fundamental features of aging, addressing both its principal targets and associated pathologies [10,15]. By integrating the above-mentioned strategies, this list reflects a multi-target, systems-level approach to geroprotection. Each activity is chosen by the evidence that it is related to longevity pathways observed in humans, ensuring relevance to both basic aging biology and age-associated diseases. This list, grouped by categories, is given below.

#### 2.1.1. Prevention of Free Radical Oxidation

Category: free radical scavengers, antioxidants; transition metal chelators; and lipid peroxidation-resistant fatty acids.

Relevant biological activities implemented in PASS GERO:Free radical scavenger;Antioxidant;Adenosine deaminase inhibitor;Lipid peroxidase inhibitor;Phosphatidic acid synthesis inhibitor;Chelator (Iron).

#### 2.1.2. Maintenance of Mitochondrial Function

Category: mitohormetics, mitochondrial electron transport chain inhibitors; mitophagy inducers; PPARγ/PGC-1α activators; and NAD+ precursors.

Relevant biological activities implemented in PASS GERO:Mitochondrial electron transport inhibitor;Peroxisome proliferator-activated receptor gamma (PPARγ) agonist.

#### 2.1.3. Proteostasis Support

Category: anti-glycation agents; crosslink inhibitors of advanced glycation end products (AGEs); RAGE receptor antagonists; anti-amyloid compounds; extracellular matrix turnover stimulators; autophagy inducers; proteasome activators; and partial translation inhibitors.

Relevant biological activities implemented in PASS GERO:RAGE receptor antagonist;Antiamyloidogenic;Autophagy inducer.

#### 2.1.4. Senolytics/Senostatics

Category: compounds promoting selective elimination of senescent cells or delaying their emergence.

Relevant biological activities implemented in PASS GERO:Senolytic activity.

#### 2.1.5. Suppression of Genomic Instability

Category: antimutagenic compounds; telomere stabilizers; and retrotransposition inhibitors.

Relevant biological activities implemented in PASS GERO:Antimutagenic;Telomerase stimulant;DNA-directed RNA polymerase inhibitor.

#### 2.1.6. Epigenetic Regulators

Category: HDAC/HAT inhibitors and SIRT activators.

Relevant biological activities implemented in PASS GERO:Histone deacetylase (HDAC) inhibitors (classes I–IV, isoforms 1–11, SIRT1–7);Sir2p inhibitor;Histone acetyltransferase (HAT) inhibitors (KAT2A/B, KAT5–7, p300, PCAF, RTT109);Nuclear receptor coactivator 1/3 inhibitors.

#### 2.1.7. Aging-Related Signaling Pathway Inhibitors

Category: mTORC1, PI3K, Ras, Myc, AT1 inhibitors; and AMPK and Klotho activators.

Relevant biological activities implemented in PASS GERO:mTOR complex 1 inhibitor;Phosphatidylinositol 3-kinase (PI3K) inhibitors (alpha, beta, delta, gamma);Myc inhibitor;Angiotensin AT1 receptor antagonist;AMP-activated protein kinase (AMPK) subunit inhibitors.

#### 2.1.8. Anti-Inflammatory Agents

Category: NF-κB inhibitors and NRF2 activators.

Relevant biological activities implemented in PASS GERO:NF-κB pathway inhibitors (NF-κB1, NF-κB2, RelA, RelB);NRF2 stimulant;COX-2, iNOS inhibitors;IGF-1–6 antagonists;HIF-1α inhibitor.

#### 2.1.9. Antifibrotic Agents

Category: TGF-β → ALK5 → p-Smad2/3 pathway inhibitors.

Relevant biological activities implemented in PASS GERO:TGF-β antagonists (isoforms 1–3);TGF-β receptor type I kinase inhibitor;SMAD3 inhibitor.

#### 2.1.10. Neurotrophic Factors

Category: BDNF mimetics and central/peripheral nervous system support agents.

Relevant biological activities implemented in PASS GERO:Neurotrophic factor enhancer;TRKB agonist.

#### 2.1.11. Barrier Function Preservation

Category: matrix metalloproteinase (MMP) inhibitors and tight junction protein synthesis activators.

Predicted Activities:Metalloproteinase-9 inhibitor.

#### 2.1.12. Immunomodulators

Category: thymus regeneration promoters and naïve T-cell pool preservers.

Relevant biological activities implemented in PASS GERO:Estradiol 17β-dehydrogenase inhibitors (isoforms 1–3);5-Alpha-reductase inhibitors (isoforms 1–2);Melatonin receptor agonists (subtypes 1–5);JAK/STAT pathway inhibitors (JAK1–3, STAT1–6).

#### 2.1.13. Miscellaneous

Category: age-related degenerations/age-associated memory impairment treatment and some antidiabetic mechanisms.

Relevant biological activities implemented in PASS GERO:5 Hydroxytryptamine 3 antagonist;5 Hydroxytryptamine 7 agonist;Age-related macular degeneration treatment;Alpha 2 adrenoreceptor antagonist;Insulin growth factor antagonist;Sodium/glucose cotransporter 2 inhibitor.

### 2.2. Implementation

Integrating this extensive, mechanistically diverse activity dataset with a large-scale training set of more than 1.48 million compounds yielded a comprehensive SAR knowledge base. As detailed in Table 1, the 117 resulting prediction models exhibit consistently high Invariant Accuracy of Prediction (IAP) scores, with a mean exceeding 0.96. These results indicate that the models are not merely descriptive but highly predictive—an essential criterion for practical use in drug discovery. Their robustness—evidenced by strong performance across categories ranging from Senolytic to Epigenetic Regulator–establishes PASS GERO as a reliable tool for generating mechanistically grounded hypotheses about compound activity, which we subsequently validated.

According to Table 1, the most frequent activities belong to the following categories: Epigenetic regulators (38 activities), Immunomodulators (19 activities), Aging-related signaling pathway inhibitors (15 activities), and Anti-inflammatory agents (14 activities). They comprise about 73% of all activities covered by the current version of PASS GERO.

The average IAP value obtained in the leave-one-out cross-validation, IAP_LOO CV_, is 0.967. The average IAP value obtained in the 20-Fold cross-validation, IAP_20-fold CV_, is 0.966.

As shown above, the IAP_LOO CV_ and IAP_20-fold CV_ values are almost equal. Thus, the PASS GERO SAR knowledgebase proves to be robust and provides both high accuracy and predictivity.

### 2.3. Web Application

We developed the web application PASS GERO https://www.way2drug.com/gero/ (accessed on 6 August 2025) that predicts probable mechanisms of anti-aging action for drug-like compounds and is freely available for academic aging-related studies. It offers predictive models to assess geroprotective activities through SAR analysis, enabling researchers to prioritize compounds with activity against aging targets like sirtuins, senescence-associated pathways, etc. It covers 117 different anti-aging activities providing the opportunity to evaluate the complex geroprotective potential for the analyzed compounds.

One may use a particular structural formula that can be drawn using Marvin JS, or loaded using a SMILES string, or, if the structure belongs to an existing drug, it can be identified by the drug name. To perform a prediction, the user may choose one of the twelve categories of anti-aging biological activities or select the option “All activities”. Another parameter that the user should select is the cutoff on Pa values (see Section 4).

Figure 2 shows the prediction results for Urolithin A (URO A), which elevated antioxidant enzymes (SOD, CAT) and reduced lipid peroxidation (MDA) in liver tissues [28], also, URO A lowered pro-inflammatory cytokines (TNF-α, IL-6, NF-κB) [28]. The results of a clinical study of URO A in humans showed that it enhanced mitochondrial activity in older human participants [29]. In non-polar media, compounds UroA exhibit antiradical capabilities against the Radical Adduct Formation (RAF) processes [30]. As shown in Figure 2, the prediction results correlate well with the experimental data, which demonstrates PASS GERO applicability for geroprotective activity assessment.

## 3. Discussion

Currently, no pharmaceutical agent is officially approved for medical use as a geroprotector. The literature analysis reveals several drugs corresponding to the criteria for classification as potential geroprotectors, some of which are being investigated in clinical trials for treating age-associated diseases [14,15,31,32,33,34,35]. Information about these drugs is provided in Table 2. To retrospectively validate the results of PASS GERO, we predicted the geroprotective activity spectra for the reference compounds. Some prediction results with the cutoff Pa > Pi are summarized in Table 2.

The full spectrum of predicted activities, along with the probabilities of being active or inactive, is provided in the Appendix A. External validation (Table 2) shows strong concordance between the known biological mechanisms of the reference compounds and the activities predicted by PASS GERO. However, this correlation should be interpreted with caution.

### 3.1. Relationship Between Mechanisms of Action and Predicted Activities

Predicted activities denote molecular mechanisms that likely contribute to the observed, higher-level mechanism of action (MoA). For example, the established senolytic MoA of Dasatinib and Navitoclax is reflected in the high-probability prediction of “Senolytic” activity. Conversely, for Resveratrol, its complex, multifaceted MoA (e.g., “improvement of mitochondrial function”) is resolved into a set of specific predicted activities—such as “Mitochondrial electron transport inhibitor”, “Histone deacetylase SIRT1 activator”, and “Autophagy inducer”—each representing a potential molecular pathway through which its overall geroprotective effect is achieved.

### 3.2. Polypharmacology and Prioritizing Predictions

Predicting multiple activities for a single compound is not only expected but also a core feature of PASS GERO, reflecting the polypharmacology of many effective geroprotectors. Not all predicted activities carry equal weight. Confidence in each prediction is quantified by the probabilities Pa (Probability of being Active) and Pi (Probability of being Inactive). A higher Pa indicates greater confidence that the compound exhibits the specified activity. Researchers should prioritize activities with Pa > Pi and, more stringently, those with a high Pa (e.g., Pa > 0.5) for further experimental investigation. The profile should be evaluated holistically: a compound with several medium-probability predictions across synergistic pathways (e.g., antioxidant, anti-inflammatory, and autophagy-inducing) may be more promising than one with a single high-probability prediction for an isolated activity.

### 3.3. Confidence, Validation, and Overprediction

While the high cross-validation accuracy (IAP ≈ 0.97) demonstrates strong internal predictive performance, not every predicted activity will be experimentally confirmed across all compounds. The model identifies potential activities based on structural similarity to known actives. Some predictions may reflect true but as-yet-untested biological effects, offering testable hypotheses for research. Others may be overpredictions—an acknowledged limitation of in silico methods. For example, the model may correctly flag a compound’s potential to interact with a target based on its structure, yet the interaction may not be physiologically relevant due to factors such as bioavailability, metabolism, or cellular context.

Accordingly, PASS GERO’s outputs are not definitive claims of activity but a ranked, probability-weighted set of testable hypotheses. The tool prioritizes the most promising candidates and identifies their most likely mechanisms of action, thereby focusing limited experimental resources. As with any computational assessment, all predicted activities require rigorous validation in vitro and in vivo. This cautious framing is essential for the tool’s effective use in geroprotector discovery.

## 4. Methods and Materials

### 4.1. The Training Set

The majority of SAR data for relevant biological activities were obtained based on PASS 2024; the data on senolytic activity was taken from the publication of Wong et al., 2023 [19]. After the training, the following statistics represent the obtained SAR knowledgebase:The number of substances in the training set: 1,483,030;The number of unique MNA descriptors: 141,153;The number of activities selected for the prediction: 117.

### 4.2. SAR Knowledgebase for Geroprotective Activities

Based on the information from the prepared set of structures with geroprotective activity, we performed the training procedure using the general machine approach realized in PASS. This approach is described in several publications earlier [47,48,49]. In brief, it used the MNA (Multilevel Neighborhoods of Atoms) descriptors reflecting the physico-chemical mechanisms of ligand-target interactions [50], and an improved naïve Bayes classifier as a machine learning approach. During the training, the leave-one-out procedure is used, which increases the predictivity of the obtained SAR models. The accuracy of each model is estimated with Invariant Accuracy of Prediction (IAP) calculated by the application of leave-one-out cross-validation; and predictivity—by the application of 20-fold cross-validation [48,49]. As a result of the training, the PASS GERO SAR knowledgebase is developed. This knowledgebase can be further applied for the prediction of geroprotective activity for new compounds using their structural formulae presented in SDF or MOL formats as an input file. An output file includes the lists of the predicted activities with two probabilities, Pa and Pi, which estimate the probability of belonging to the active and inactive compounds, respectively. The activities in this list are arranged in descending order of ΔP = Pa − Pi values, where higher ΔP values indicate a greater confidence in the prediction. By default, all activities predicted with ΔP > 0 are considered probable; however, depending on the particular task, the user may select another cutoff. Additional recommendations for the interpretation of the prediction results are presented in our publications [47,48,49].

### 4.3. Web Application

To provide the possibility for a wide utilization of the PASS GERO software by the scientific community, we developed a freely available web application at the Way2Drug portal https://www.way2drug.com/gero/ (accessed on 6 August 2025). The PHP programming languages were used to develop the server side of the web application. HTML, CSS, PHP, and JavaScript were used to create the web interface.

## 5. Conclusions

Despite recent progress, in silico discovery of geroprotectors faces substantial obstacles. The pronounced chemical heterogeneity of candidate compounds—from small synthetic molecules to complex natural products—complicates the construction of robust training sets. The field also lacks reliable, validated biomarkers to quantify the effects of aging interventions, a gap further compounded by the multifactorial nature of aging itself. These scientific challenges are exacerbated by ethical considerations and the formidable complexity of clinical trial design for aging interventions, collectively hindering clinical translation.

Our study directly addresses the first of these challenges. PASS GERO is specifically designed to navigate the chemical diversity of candidate geroprotectors. By leveraging MNA descriptors and a large, purposefully curated training set spanning a broad range of chemical scaffolds, our model provides a unified framework for evaluating structurally diverse compounds. Validation results indicate strong predictive performance for established geroprotectors with markedly different structures—from the complex macrolide rapamycin to the simple polyphenol resveratrol—underscoring its utility. While it does not solve the biomarker validation challenge, PASS GERO offers a systematic, mechanism-based prioritization strategy to guide preclinical research toward the most promising multi-target candidates, thereby de-risking the earliest stages of drug discovery in geroscience.

Critically, validation of PASS GERO against a panel of established geroprotectors confirms its predictive power and biological relevance. The software accurately recapitulates the known polypharmacological profiles of reference compounds.

This capability to generate a comprehensive mechanistic profile for a single molecule—whether from a hand-drawn structure or a SMILES string—provides researchers with a powerful tool to prioritize candidates for virtual screening, inform the synthesis of novel analogs, and design targeted experimental studies, thereby de-risking the early stages of geroprotector discovery.

Nevertheless, significant challenges remain—chief among them the lack of validated biomarkers of aging and the complexity of designing clinical trials for aging interventions. Although PASS GERO cannot resolve these translational hurdles, it directly addresses the initial bottleneck: compound selection. Future work will expand the model’s scope to encompass emerging mechanisms, including hormetins and gut microbiota modulators. As this integrated knowledge base grows, PASS GERO will gain predictive power, improving the likelihood of identifying clinical candidates that target the hallmarks of aging and prevent age-related diseases.

## Figures and Tables

**Figure 1 ijms-26-08858-f001:**
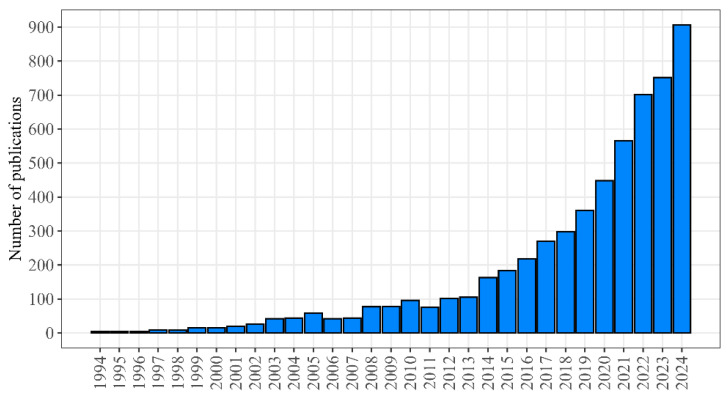
The number of publications in the field of geroprotection included in the PubMed database (PubMed query: “geroprotector OR anti-aging medicine”).

**Figure 2 ijms-26-08858-f002:**
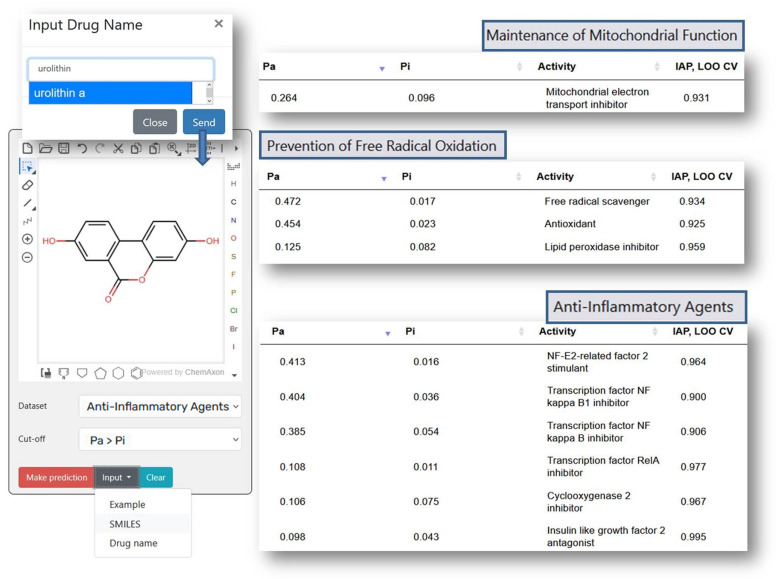
The prediction results for Urolithin A with PASS GERO web application.

**Table 1 ijms-26-08858-t001:** Characteristics of classification models for assessing geroprotective activities.

N *	IAP **	Geroprotective Activity
**Category**		**Prevention of Free Radical Oxidation**
2641	0.934	Free radical scavenger
4991	0.925	Antioxidant
315	0.988	Adenosine deaminase inhibitor
1153	0.959	Lipid peroxidase inhibitor
799	0.967	Chelator
45	0.986	Chelator, Iron
**Category**		**Maintenance of Mitochondrial Function**
4	0.931	Mitochondrial electron transport inhibitor
3444	0.986	Peroxisome proliferator-activated receptor gamma agonist
**Category**		**Proteostasis Support**
56	0.919	RAGE receptor antagonist
3286	0.931	Antiamyloidogenic
500	0.836	Autophagy inducer
**Category**		**Senolytics/senostatics**
44	0.897	Senolytic
**Category**		**Suppression of Genomic Instability**
40	0.974	Antimutagenic
11	0.999	Telomerase stimulant
3232	0.967	DNA-directed RNA polymerase inhibitor
**Category**		**Epigenetic Regulators**
4997	0.993	Histone deacetylase 1 inhibitor
2138	0.992	Histone deacetylase 2 inhibitor
1744	0.996	Histone deacetylase 3 inhibitor
1110	0.994	Histone deacetylase 4 inhibitor
1879	0.993	Histone deacetylase 5 inhibitor
3455	0.994	Histone deacetylase 6 inhibitor
533	0.992	Histone deacetylase 7 inhibitor
1637	0.992	Histone deacetylase 8 inhibitor
501	0.990	Histone deacetylase 9 inhibitor
581	0.995	Histone deacetylase 10 inhibitor
656	0.994	Histone deacetylase 11 inhibitor
8	1.000	Histone deacetylase 1A inhibitor
42	1.000	Histone deacetylase 1B inhibitor
337	0.956	Histone deacetylase SIRT1 inhibitor
311	0.981	Histone deacetylase SIRT1 stimulant
586	0.986	Histone deacetylase SIRT2 inhibitor
5	0.999	Histone deacetylase SIRT2 stimulant
161	0.982	Histone deacetylase SIRT3 inhibitor
68	0.995	Histone deacetylase SIRT5 inhibitor
14	0.885	Histone deacetylase SIRT6 inhibitor
6219	0.992	Histone deacetylase class I inhibitor
5008	0.993	Histone deacetylase class II inhibitor
2566	0.993	Histone deacetylase class IIa inhibitor
3485	0.994	Histone deacetylase class IIb inhibitor
803	0.968	Histone deacetylase class III inhibitor
656	0.994	Histone deacetylase class IV inhibitor
10,985	0.984	Histone deacetylase inhibitor
35	0.955	Histone acetyltransferase KAT2A inhibitor
97	0.990	Histone acetyltransferase KAT2B inhibitor
21	0.965	Histone acetyltransferase KAT5 inhibitor
243	0.999	Histone acetyltransferase KAT6A inhibitor
11	0.980	Histone acetyltransferase KAT7 inhibitor
41	0.973	Histone acetyltransferase PCAF inhibitor
84	0.989	Histone acetyltransferase RTT109 inhibitor
1100	0.935	Histone acetyltransferase inhibitor
399	0.974	Histone acetyltransferase p300 inhibitor
115	0.891	Nuclear receptor coactivator 1 inhibitor
171	0.840	Nuclear receptor coactivator 3 inhibitor
**Category**		**Aging-related Signaling Pathway Inhibitors**
310	0.964	mTOR complex 1 inhibitor
11,736	0.989	Phosphatidylinositol 3-kinase alpha inhibitor
2922	0.988	Phosphatidylinositol 3-kinase beta inhibitor
4994	0.989	Phosphatidylinositol 3-kinase delta inhibitor
6895	0.986	Phosphatidylinositol 3-kinase gamma inhibitor
18,093	0.984	Phosphatidylinositol 3-kinase inhibitor
507	0.986	Myc inhibitor
6176	0.994	Angiotensin AT1 receptor antagonist
1625	0.998	Angiotensin AT1A receptor antagonist
1413	0.999	Angiotensin AT1B receptor antagonist
394	0.934	AMP-activated protein kinase inhibitor
311	0.930	AMP-activated protein kinase, alpha-1 subunit inhibitor
43	0.955	AMP-activated protein kinase, alpha-2 subunit inhibitor
18	0.892	AMP-activated protein kinase, beta-1 subunit inhibitor
15	0.945	AMP-activated protein kinase, gamma-1 subunit inhibitor
**Category**		**Anti-Inflammatory Agents**
1787	0.906	Transcription factor NF kappa B inhibitor
483	0.900	Transcription factor NF kappa B1 inhibitor
74	0.995	Transcription factor NF kappa B2 inhibitor
99	0.977	Transcription factor RelA inhibitor
10	0.964	NF-E2-related factor 2 stimulant
4885	0.992	Poly(ADP-ribose) polymerase 1 inhibitor
1699	0.980	Inducible nitric oxide synthase inhibitor
7079	0.967	Cyclooxygenase 2 inhibitor
2327	0.960	Ribosomal protein S6 kinase inhibitor
3898	0.979	Insulin growth factor antagonist
3854	0.979	Insulin like growth factor 1 antagonist
3	0.995	Insulin like growth factor 2 antagonist
44	1.000	Insulin like growth factor 3 antagonist
2125	0.913	Hypoxia-inducible factor 1-alpha inhibitor
**Category**		**Antifibrotic Agents**
4088	0.929	Fibrosis treatment
2115	0.988	Transforming growth factor beta 1 antagonist
182	0.991	Transforming growth factor beta 2 antagonist
75	0.965	Transforming growth factor beta 3 antagonist
2367	0.980	Transforming growth factor beta antagonist
210	0.996	TGF beta receptor type I kinase inhibitor
29	0.857	SMAD3 inhibitor
**Category**		**Neurotrophic Factors**
524	0.963	Neurotrophic factor
77	0.934	Neurotrophic factor enhancer
59	0.974	TRKB agonist
**Category**		**Barrier Function Preservation**
2975	0.980	Metalloproteinase-9 inhibitor
**Category**		**Immunomodulators**
704	0.997	Estradiol 17 beta-dehydrogenase 1 inhibitor
579	0.994	Estradiol 17 beta-dehydrogenase 2 inhibitor
302	0.996	Estradiol 17 beta-dehydrogenase 3 inhibitor
1232	0.994	Estradiol 17 beta-dehydrogenase inhibitor
687	0.996	5-Alpha-reductase 1 inhibitor
642	0.994	5-Alpha-reductase 2 inhibitor
1738	0.994	5-Alpha-reductase inhibitor
372	0.995	Melatonin receptor 1 agonist
188	0.997	Melatonin receptor 2 agonist
615	0.995	Melatonin receptor agonist
6346	0.992	Janus tyrosine kinase 1 inhibitor
10,871	0.980	Janus tyrosine kinase 2 inhibitor
8003	0.983	Janus tyrosine kinase 3 inhibitor
15,102	0.978	Janus tyrosine kinase inhibitor
1174	0.918	Transcription factor STAT inhibitor
26	0.831	Transcription factor STAT1 inhibitor
881	0.925	Transcription factor STAT3 inhibitor
47	0.874	Transcription factor STAT5 inhibitor
247	0.957	Transcription factor STAT6 inhibitor
**Category**		**Miscellaneous**
2435	0.988	5 Hydroxytryptamine 3 antagonist
91	0.986	5 Hydroxytryptamine 7 agonist
116	0.916	Age-related macular degeneration treatment
2772	0.975	Alpha 2 adrenoreceptor antagonist
2226	0.998	Sodium/glucose cotransporter 2 inhibitor

* N is the number of active compounds in the training set; ** IAP is the Invariant Accuracy of Prediction for each activity estimated by the leave-one-out cross-validation.

**Table 2 ijms-26-08858-t002:** Potential geroprotectors under investigation and their predicted activities.

Name and Structure	MoA *	Predicted Activities
Atorvastatin 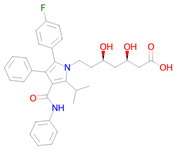	Targeting senescent cells, inhibition of CDKN2A expression [36]	Mitochondrial electron transport inhibitor Senolytic
Acetylsalicylic acid 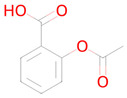	Targeting senescent cells, inhibition of CDKN2A expression [36]	Transcription factor STAT1 inhibitor Transcription factor NF kappa B inhibitor SMAD3 inhibitor Autophagy inducer Transcription factor NF kappa B1 inhibitor Free radical scavenger Transcription factor STAT inhibitor NF-E2-related factor 2 stimulant Antioxidant Antimutagenic Histone deacetylase SIRT6 inhibitor Neurotrophic factor enhancer Hypoxia-inducible factor 1 alpha inhibitor Chelator
Vildagliptin 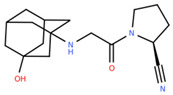	Reduction in stress-induced cellular senescence rate [37]	Senolytic
Dasatinib 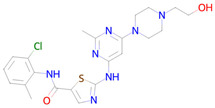	Targeting senescent cells, inhibition of PI3K (Phosphoinositide 3-kinase) [14]	Senolytic
Quercetin 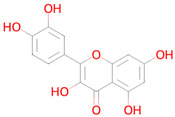	Targeting senescent cells, inhibition of PI3K (Phosphoinositide 3-kinase) [14]	Senolytic
Curcumin 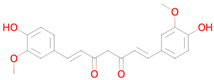	Inhibitor of DNMT1, DNMT3B, HATs, HDAC2, and HDAC8 [38]	Free radical scavenger Antioxidant Transcription factor NF kappa B inhibitor Antimutagenic Antiamyloidogenic Transcription factor NF kappa B1 inhibitor Autophagy inducer SMAD3 inhibitor Transcription factor STAT inhibitor Hypoxia-inducible factor 1 alpha inhibitor Histone acetyltransferase inhibitor Nuclear receptor coactivator 3 inhibitor Lipid peroxidase inhibitor Chelator NF-E2-related factor 2 stimulant Mitochondrial electron transport inhibitor Insulin like growth factor 2 antagonist Nuclear receptor coactivator 1 inhibitor
Metformin 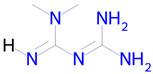	Autophagy, mitochondrial biogenesis [14]	Histone deacetylase SIRT6 inhibitor Transcription factor NF kappa B1 inhibitor NF-E2-related factor 2 stimulant
Navitoclax 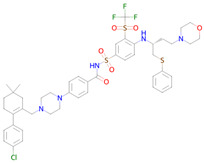	Targeting senescent cells, inhibition of Bcl-2 and Bcl-xl [39]	Senolytic Transcription factor STAT5 inhibitor
Nicotinamide riboside 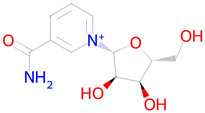	Replenishment of NAD+ deficiency [40]	Autophagy inducer Nuclear receptor coactivator 3 inhibitor Transcription factor NF kappa inhibitor Histone deacetylase SIRT1 stimulant Transcription factor STAT inhibitor Mitochondrial electron transport inhibitor
Rapamycin (Sirolimus) 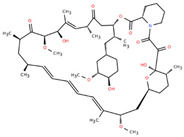	Targeting senescent cells, mTOR inhibitor [14]	Senolytic mTOR complex 1 inhibitor Transcription factor NF kappa B inhibitor Neurotrophic factor Fibrosis treatment Hypoxia-inducible factor 1 alpha inhibitor
Resveratrol 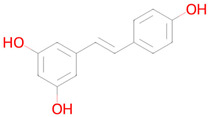	Reduction in oxidative stress, anti-inflammatory effect, improvement of mitochondrial function, regulation of apoptosis [41]	Autophagy inducer Antimutagenic Free radical scavenger Antioxidant SMAD3 inhibitor Nuclear receptor coactivator 3 inhibitor Antiamyloidogenic Transcription factor NF kappa B inhibitor Histone deacetylase SIRT2 stimulant Transcription factor NF kappa B1 inhibitor NF-E2-related factor 2 stimulant Chelator Lipid peroxidase inhibitor Histone deacetylase SIRT1 stimulant Transcription factor STAT inhibitor Hypoxia-inducible factor 1 alpha inhibitor Nuclear receptor coactivator 1 inhibitor Neurotrophic factor enhancer Insulin like growth factor 2 antagonist Mitochondrial electron transport inhibitor Histone acetyltransferase inhibitor Transcription factor RelA inhibitor Senolytic
Spermidine 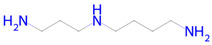	Autophagy inducer [42]	Autophagy inducer Antimutagenic Histone deacetylase SIRT6 inhibitor Chelator Transcription factor NF kappa B1 inhibitor NF-E2-related factor 2 stimulant Antiamyloidogenic Transcription factor STAT1 inhibitor Histone deacetylase SIRT1 inhibitor Inducible nitric-oxide synthase inhibitor SMAD3 inhibitor Nuclear receptor coactivator 1 inhibitor Transcription factor STAT inhibitor Nuclear receptor coactivator 3 inhibitor
Sulforophane 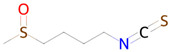	Antioxidant and anti-inflammatory effects, proteasome activation [43]	NF-E2-related factor 2 stimulant Transcription factor STAT1 inhibitor Inducible nitric-oxide synthase inhibitor Antioxidant Antiamyloidogenic
Torin 2 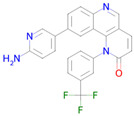	Inhibitor of TORC1 and TORC2 [44]	Transcription factor STAT1 inhibitor Fibrosis treatment AMP-activated protein kinase inhibitor mTOR complex 1 inhibitor
Urolithin A 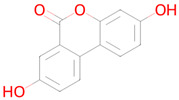	Mitophagy [45]	Antimutagenic SMAD3 inhibitor RAGE receptor antagonist Nuclear receptor coactivator 3 inhibitor Free radical scavenger Antioxidant Transcription factor NF kappa B inhibitor NF-E2-related factor 2 stimulant Antiamyloidogenic Autophagy inducer Neurotrophic factor enhancer Hypoxia-inducible factor 1 alpha inhibitor Nuclear receptor coactivator 1 inhibitor Transcription factor STAT inhibitor Mitochondrial electron transport inhibitor 17 beta-dehydrogenase 3 inhibitor Senolytic
Fisetin 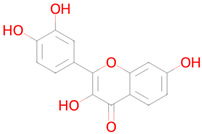	Targeting senescent cells, inhibition of PI3K (Phosphoinositide 3-kinase) [46]	Antimutagenic Free radical scavenger Antioxidant SMAD3 inhibitor Autophagy inducer TRKB agonist Nuclear receptor coactivator 3 inhibitor Transcription factor NF kappa B inhibitor Hypoxia-inducible factor 1 alpha inhibitor NF-E2-related factor 2 stimulant Transcription factor NF kappa B1 inhibitor Lipid peroxidase inhibitor Mitochondrial electron transport inhibitor Histone deacetylase SIRT2 stimulant Antiamyloidogenic Transcription factor STAT inhibitor Chelator Histone deacetylase SIRT1 stimulant RAGE receptor antagonist Estradiol 17 beta-dehydrogenase inhibitor Neurotrophic factor enhancer

* MoA—Mechanism of Action.

## Data Availability

We provide full academic access to the PASS GERO web resource. All senolytics-related data utilized in this study are derived from publicly accessible sources. Specifically, these datasets are available in the Appendix A of our cited reference https://doi.org/10.1038/s43587-023-00415-z (accessed on 6 August 2025), requiring no additional deposition. Using PASS GERO for prediction is free.

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
