# Peer review of "In Silico Assessment of Potential Geroprotectors: From Separate Endpoints to Complex Pharmacotherapeutic Effects"

_ijms, 2025, doi:10.3390/ijms26188858_

Round 1
Reviewer 1 Report
Comments and Suggestions for Authors
The work is very interesting, it deals with very interesting and current issues.
Unfortunately, the abstract and the conclusion are extremely general, and I think that their correction is needed.
Certain changes are required before acceptance.
1. In line 25, no citation is needed
2. The abstract is too general
3. Lines 103-107 you only state the aim, you need to define it clearly, and not you, but you also present the results to us
4. Explain the principle on which the PASS GERO web application works in the introduction.
5. The conclusion is too general
Author Response
We thank the reviewer for careful review and helpful comments.
Comments 1:
In line 25, no citation is needed.
Response 1:
There is no citation now, moreover we enhanced the abstract considering the max volume of 200.
Comments 2:
The abstract is too general
Response 2:
See answer 1.
Comments 3:
Lines 103-107 you only state the aim, you need to define it clearly, and not you, but you also present the results to us
Response 3:
We enlarged the Introduction section according to your comments and clarified the principles of PASS GERO.
Comments 4:
Explain the principle on which the PASS GERO web application works in the introduction.
Response 4:
See answer 3.
Comments 5:
The conclusion is too general
Response 5:
The conclusion was indeed too general; now it is significantly refined to show all aspects of work, benefits from PASS GERO, its limitations, plans and overall problems in this area.
Reviewer 2 Report
Comments and Suggestions for Authors
In this manuscript, the authors presented an aging related predictive model along with its web server implementation. This is an important extrapolation of PASS model specially for aging related mechanism and activity. I recommend publishing this manuscript once the authors address the following comments which would make the manuscript and the study even more robust.
- Even thought the manuscript mentions it uses the previous PASS model to build upon it, I think more discussion on the machine learning model used here is important such as how the model was trained with the new sets of data. I think sharing the code through GitHub will help the community and the readers to understand more.
- In table 2, I would recommend adding a blank line after each compound.
- Could the authors comment on how the Mechanism of Action and the predicted activities in table 2 related? Also, there are several predicted activities for many of the candidates- are they equally important and how confident the user can be with these different predictions?
Author Response
We thank the reviewer for careful review and helpful comments.
Comments 1:
Even thought the manuscript mentions it uses the previous PASS model to build upon it, I think more discussion on the machine learning model used here is important such as how the model was trained with the new sets of data. I think sharing the code through GitHub will help the community and the readers to understand more.
Response 1:
PASS GERO is based on the PASS approach, which has been under development since the early 1990s. It has been described in numerous publications (e.g., refs. [47–49] in this manuscript). Because this study was supported by the Russian Ministry of Science and Higher Education (see the Acknowledgements), our contract with the Ministry does not permit us to share the code on GitHub until the Russian Patent Agency issues the relevant Certificate (expected in 2026). Since the problem of discovering and developing new geroprotectors is extremely important, it is necessary to present to the Scientific Community the PASS GERO web service as soon as possible, which enables the in silico evaluation of more than 100 types of anti-aging activity. This service is freely available to academic researchers (no registration required), which will significantly accelerate the necessary developments in this field.
Comments 2:
In table 2, I would recommend adding a blank line after each compound.
Response 2:
Done.
Comments 3:
Could the authors comment on how the Mechanism of Action and the predicted activities in table 2 related? Also, there are several predicted activities for many of the candidates- are they equally important and how confident the user can be with these different predictions?
Response 3:
Yes, we enriched Results, Discussion and Materials/Methods with additional comments about models’ interpretability, confidence of the prediction, link of mechanisms of action and prediction results. We do hope each step is clearer now.
Round 2
Reviewer 2 Report
Comments and Suggestions for Authors
I would like to thank the author for addressing my comments. I recommend publishing this manuscript.